# Impact of Formulation Conditions on Lipid Nanoparticle Characteristics and Functional Delivery of CRISPR RNP for Gene Knock-Out and Correction

**DOI:** 10.3390/pharmaceutics14010213

**Published:** 2022-01-17

**Authors:** Johanna Walther, Danny Wilbie, Vincent S. J. Tissingh, Mert Öktem, Heleen van der Veen, Bo Lou, Enrico Mastrobattista

**Affiliations:** Department of Pharmaceutics, Utrecht Institute of Pharmaceutical Sciences (UIPS), Utrecht University, Universiteitsweg 99, 3584 CG Utrecht, The Netherlands; j.walther@uu.nl (J.W.); d.wilbie@uu.nl (D.W.); vincenttissingh@gmail.com (V.S.J.T.); m.oktem@uu.nl (M.Ö.); h.vanderveen@students.uu.nl (H.v.d.V.); bo.lou@outlook.com (B.L.)

**Keywords:** CRISPR, LNP, formulation, NHEJ, HDR, AF4, delivery

## Abstract

The CRISPR-Cas9 system is an emerging therapeutic tool with the potential to correct diverse genetic disorders. However, for gene therapy applications, an efficient delivery vehicle is required, capable of delivering the CRISPR-Cas9 components into the cytosol of the intended target cell population. In this study, we optimized the formulation conditions of lipid nanoparticles (LNP) for delivery of ready-made CRISPR-Cas9 ribonucleic protein (RNP). The buffer composition during complexation and relative DOTAP concentrations were varied for LNP encapsulating in-house produced Cas9 RNP alone or Cas9 RNP with additional template DNA for gene correction. The LNP were characterized for size, surface charge, and plasma interaction through asymmetric flow field flow fractionation (AF4). Particles were functionally screened on fluorescent reporter cell lines for gene knock-out and gene correction. This revealed incompatibility of RNP with citrate buffer and PBS. We demonstrated that LNP for gene knock-out did not necessarily require DOTAP, while LNP for gene correction were only active with a low concentration of DOTAP. The AF4 studies additionally revealed that LNP interact with plasma, however, remain stable, whereby HDR template seems to favor stability of LNP. Under optimal formulation conditions, we achieved gene knock-out and gene correction efficiencies as high as 80% and 20%, respectively, at nanomolar concentrations of the CRISPR-Cas9 RNP.

## 1. Introduction

The clustered regularly interspaced short palindromic repeats (CRISPR) associated (Cas) endonuclease proteins, such as Cas9, have emerged in recent years as a viable therapeutic option for genetic diseases. The Cas9 endonuclease was first identified as a bacterial defense mechanism against viral infections and has been repurposed into a powerful tool to cleave DNA in an RNA-guided fashion in various cell types. The Cas9 protein, together with a guide RNA molecule, forms an active ribonucleoprotein (RNP) complex [1]. DNA cleavage is mediated by recognition of a 20-nucleotide sequence between the guide RNA and the host DNA, which hybridizes and allow the nuclease to attach to its DNA target. Additionally, the presence of a protospacer-adjacent motif in the host DNA is necessary to facilitate the conformational change in the nuclease to introduce a double strand break in its target [2]. When the genomic DNA is cleaved by the Cas9 enzyme, the host DNA-damage repair response is activated [3]. In mammalian cells, the most prominent pathways are the canonical non-homologous end-joining (c-NHEJ) pathway, the microhomology-mediated end joining (MMEJ) pathway, and homology-directed repair (HDR) [4]. C-NHEJ and MMEJ are notably error-prone repair mechanisms, both of which can lead to formation of small insertions and deletions in the target gene. This, in turn, may lead to gene knock-out, which is therapeutically relevant for gene therapy of diseases caused by gain-of-function mutations [5,6,7]. HDR is mostly active in the G2/S phases of mitosis in dividing cells, and in the presence of a homologous DNA template, this pathway can lead to precise DNA repair of disrupted genes [8]. Especially, the latter signifies potential for gene therapy, thereby curing diseases by editing and correcting the genetic mutations.

Direct in vivo gene editing requires the delivery of the CRISPR-Cas9 components into the correct target cells’ nuclei [9]. SpCas9, a Cas9 protein derived from *Streptococcus pyogenes*, is currently under clinical investigation for both ex vivo and direct in vivo therapeutic applications [10,11,12]. Examples include subretinal injection of adeno-associated viral vectors encoding the CRISPR-Cas9 components for the treatment of Leber congenital amaurosis, and delivery of CRISPR-Cas9 with non-viral particles such as NTLA-2001 for targeted gene editing of hepatocytes for hereditary amyloid transthyretin amyloidosis [13,14]. Lipid nanoparticles (LNP), which employ cationic or ionizable cationic lipids, serve as particularly promising candidates for delivery of the different cargo formats of the CRISPR-Cas9 components. Since LNP complex their cargo via electrostatic interactions, they are especially suited to formulate polyanionic DNA or RNA molecules, due to their anionic phosphate backbone. However, the preassembled RNP complex, with or without co-entrapment of a DNA template to drive homology-directed repair, can also be formulated in LNPs, as was recently demonstrated [13,15,16,17].

Direct delivery of the pre-assembled RNP has several advantages over Cas9 expressed from DNA or mRNA templates. Since RNP are pre-assembled, they are directly active once inside the nuclei of target cells as opposed to Cas9 expression from DNA or mRNA templates. First, these need to be translated into the endonuclease in the cytosol, and subsequently, find an intact single guide RNA (sgRNA) within the cell in order to become active [18]. Related to this, direct delivery of RNP assures optimal stoichiometry between Cas9 and sgRNA and protects the sgRNA from rapid degradation within the cell [19]. Finally, RNP are short lived inside cells, with a half-life of approximately one day [20]. This limits the likelihood of off-target gene editing which has been shown to be time dependent [21,22].

Despite these advantages, delivery of RNP has met with several pharmaceutical challenges. The stability of RNP during LNP formulation is an issue. Solely relying on ionizable cationic lipids to mediate electrostatic interactions with the net negatively charged RNP requires an acidic environment. Acidic conditions can however affect RNP stability [23,24]. Therefore, in this study, formulations already used for siRNA or mRNA delivery with C12-200 ionizable lipid were further developed for delivery of RNP [23,25]. Specifically, formulation conditions must be optimized to find a good balance between RNP functionality, protection from premature clearance, and timely intracellular release. This work sought to explore several of such often overlooked steps in the pharmaceutical formulation of RNP into LNP, which, as shown here, are often critical in determining gene editing efficiency [23]. This includes buffer composition during formulation, as well as lipid composition of LNP for delivering RNP with or without a single stranded DNA (ssDNA) HDR templates. To understand the effects of these parameters, these LNP were characterized based on their size, surface charge, RNP complexation, and activity. Additionally, their stability in human plasma was studied. Lipid nanoparticles complexing RNP and HDR template were investigated on gene editing capacity in fluorescent reporter cell lines suited to read out gene knock-out and specific gene correction, resulting in promising results for in vivo gene correction.

## 2. Materials and Methods

### 2.1. General Reagents

All reagents and chemicals were acquired from Sigma-Aldrich (Zwijndrecht, The Netherlands) unless otherwise specified. 2′ O-methyl and phosphorothioate end-modified sgRNA and template DNA sequences were acquired from Sigma-Aldrich (Haverhill, the United Kingdom, sequences given in Appendix A) and stored in RNAse-free Tris EDTA-buffer pH 7.0 (Thermo Fisher, Landsmeer, The Netherlands). Primers for polymerase chain reaction (PCR) were acquired from Integrated DNA Technologies (IDT, Leuven, Belgium), sequence shown in Appendix A. In addition, 1,1′-((2-(4-(2-((2-(bis(2-hydroxydodecyl)amino)ethyl)(2-hydroxydodecyl)amino)ethyl)piperazin-1-yl)ethyl)azanediyl)bis(dodecan-2-ol) (C12-200) [25] was acquired from CordonPharma (Plankstadt, Germany), 1,2-dioleoyl-sn-glycero-3-phosphoethanolamine (DOPE) from Lipoid (Steinhausen, Switzerland), Cholesterol and 1,2-dimyristoyl-rac-glycero-3-methoxypolyethylene glycol-2000 (PEG-DMG) from Sigma-Aldrich (Zwijndrecht, The Netherlands), and 1,2-dioleoyl-3-trimethylammonium-propane (DOTAP) from Merck (Darmstadt, Germany).

#### 2.1.1. SpCas9 Protein Production and Purification

SpCas9 with a nuclear localization signal (NLS) was expressed in the LPS-free Clearcoli™ BL21 strain (Lucigen Corporation, Middleton, WI, USA) using pET15_SpCas9_NLS_His plasmid (Addgene #62731) [26]. After growth in LB-Miller medium until the OD600 reached 0.55–0.7, protein production was induced with 0.5 mm isopropyl β-d-1-thiogalactopyranoside (IPTG), followed by overnight fermentation at 18 °C. All bacteria were subsequently pelleted by centrifugation and lysed by tip sonication using a 3 mm tip (Bandelin electronic GmbH & Co. KG, Berlin, Germany), in 50 mL of phosphate buffered saline containing 25 mm imidazole on ice. The lysate was subsequently centrifuged, resuspended in the same buffer, and filtered through a 0.45 μM MiniSart filter (Sartorius, Amersfoort, The Netherlands). Immobilized metal affinity chromatography (IMAC) was performed on this lysate using a 1 mL nickel HisTrap HP column (Cytiva, Medemblik, The Netherlands) in combination with the Äkta PURE chromatography system (Cytiva, Medemblik, The Netherlands). A stepwise gradient of imidazole was applied from 25 mM, going up to 100 mM and ending at 250 mM.

After collection of all fractions, the eluted SpCas9 was dialyzed twice against storage buffer (final composition of 300 mM NaCl, 0.1 mM EDTA, 10 mm Tris, pH 7.4) at a 1:1000 ratio of sample to dialysate, followed by addition of 8.3% (*w*/*v*) glycerol prior to freezing. The samples were snap-frozen in liquid nitrogen and stored at −80 °C after dialysis.

#### 2.1.2. SpCas9 Characterization and Stability Study

The protein size and protein impurities were assessed using sodium dodecyl sulfate polyacrylamide gel electrophoresis (SDS-PAGE). The samples were treated with Laemmli sample buffer containing 12.5 mM dithiothreitol (DTT). The proteins were separated on 4–12% Bis-Tris gel (Thermo Fisher, Landsmeer, The Netherlands), after which staining was done using the Pierce silver stain kit (Fischer Scientific, Landsmeer, The Netherlands). Gels were imaged in the ChemiDoc Imaging System (Bio-Rad Laboratories B.V, Veenendaal, The Netherlands). The intensity of the gel bands was quantified by densitometry in ImageJ (version 1.52p), to calculate the protein impurities in the SpCas9 samples over time [27]. This assay was repeated periodically to determine the protein stability during 6 months of storage.

To visualize in vitro cleaving activity of SpCas9, an in-house optimized activity assay was performed. SpCas9 was first incubated with sgRNA specific for the EGFP gene (Appendix A) for 10 min at room temperature, at a molar ratio of 1:1 at a concentration of 1 μM. Subsequently, 2 μL of this RNP was mixed with 3 μL Buffer 3.1 10×, (New England Biolabs, Ipswich, MA, USA), 250 ng linearized plasmid DNA containing the enhanced green fluorescent protein (EGFP) locus (pMJ922, Addgene #78312 [28]), 1 μL Ribolock R1 RNAse inhibitor (Thermo Fisher, Landsmeer, The Netherlands) and filled to 30 μL with nuclease-free water (Thermo Scientific, Landsmeer, The Netherlands). The reaction was completed in 2 h at 37 °C. The samples were treated with 1 μL proteinase K (Thermo Fisher, Landsmeer, The Netherlands) and filled to 30 μL with nuclease-free water (Thermo Scientific, Landsmeer, The Netherlands), and then separated using agarose gel (1%) electrophoresis and visualized with 5 μL Midori Green (Nippon Genetics, Düren, Germany) staining per 100 mL of agarose. SpCas9 activity was calculated by gel densitometry, by determining the area under the curve in ImageJ, and calculating the relative cleaved fraction. This was repeated over the course of one year to determine the protein stability in storage.

### 2.2. Lipid Nanoparticle Formulation

To formulate LNP for gene knock-out (LNP-RNP), sgRNA and SpCas9 were mixed at a 1:1 molar ratio in different formulation buffers (100 mM citrate buffer (pH 4.0), Dulbecco’s phosphate buffered saline (PBS) (pH 7.4), 50 mm HEPES buffer (pH 7.4, LNP-RNP [HEPES]), or nuclease-free water at an RNP concentration of 0.4 μM. Complexation was performed for 15 min at room temperature. Concurrently, the lipids were mixed in ethanol to achieve a total lipid to sgRNA ratio of 40:1 (*w*/*w*), resulting in a total lipid weight of 9.6 μg [23]. The lipid components were C12-200, DOPE, cholesterol, PEG-DMG and DOTAP (molar ratio 35:16:46.5:2.5:variable). Different molar ratios of DOTAP were tested to find the optimal amount for complexation with RNP. The RNP and lipids were mixed by pipetting at a volume ratio of 3:1 (18 μL RNP to 6 μL lipids) and incubating for 15 min at room temperature. Subsequently, the formulation was diluted 4 times with PBS to a final RNP molar concentration of 76.9 nM in 100 μL. The formulation steps with exact volumes are shown in Appendix A.

LNP carrying RNP and HDR template (LNP-RNP-HDR) were formulated in the same manner in HEPES buffer or nuclease-free water (LNP-RNP-HDR [HEPES] and LNP-RNP-HDR [H_2_O], respectively), except that the HDR template was added at varying molar ratios of RNP/HDR template (1:2, 1:3.8, 1:5, 1:10 and 1:20) to the RNP complex, prior to complexation with the lipids.

### 2.3. Physical Characterization of Lipid Nanoparticles

LNP were diluted 1.3 times further in 1 × PBS (pH 7.4) for characterization of size and polydispersity index (PDI) through dynamic light scattering (DLS) using a Zetasizer Nano S (Malvern ALV CGS-3, Malvern, UK) (settings: temperature 25 °C, viscosity 0.8872 cP, RI 1.330). The ζ-potential was determined with a Zetasizer Nano Z (Malvern ALV CGS-3, Malvern, UK) after 9 × dilution in 10 mM HEPES buffer at pH 7.4 (settings: temperature 25 °C, viscosity 0.8872 cP, RI 1.330, dielectric constant 78.5). Each sample was measured in triplicate to determine size and ζ-potential two days after formulation.

### 2.4. Quantification of RNP Complexed with LNP

Complexation efficiencies were determined in LNP prepared in the different formulation conditions. RNP at 1.25 μM and a final formulation volume of 0.47 mL in PBS were used. For determination of SpCas9 complexation, the LNP formulation was additionally dialyzed against 1 × HEPES buffered saline (HBS) with Float-A-Lyzer molecular weight cut-off (MWCO) 300 kDa dialysis chambers (Avantor^®^, Arnhem, The Netherlands) to remove free SpCas9 from the formulation.

Reversed-phase high performance liquid chromatography (HPLC) (Waters Alliance e2695, Milford, MA, USA) was performed to determine the amount of SpCas9 that was complexed with LNP, using an Xbridge protein BEH C4 300 Å column (Waters #186004505) with a linear acetonitrile gradient, from 5% to 100% in 5 min and back again in 1 min, with 10 min of total elution time. The mobile phase additionally contained 0.1% trifluoroacetic acid. The column was heated at 30 °C. Fluorescence detection was set at ex. 280 nm, em. 350 nm (10 pts/s), and the UV-Vis detection was set at 214 and 280 nm (2 pts/s). Samples were treated with 2% Triton X-100 for 5 min before injection. Samples were injected with an injection volume of 50 μL at a flow rate of 1 mL/min. A calibration curve of empty LNP spiked with SpCas9, with a concentration range of 0–300 nM and treated with 2% Triton X-100, was used to quantify the SpCas9 concentration.

The Quant-iTTM RiboGreen^®^ RNA kit (Fisher Scientific, Landsmeer, The Netherlands) was used to determine the complexation efficiency of sgRNA. The protocol provided by the supplier was followed, except that sgRNA was used instead of the RNA standard to generate a calibration curve in RNAse-free TE buffer. A calibration curve with and without 2% Triton X-100 was made in duplicate. LNP samples and the calibration curve that were not treated with 2% Triton X-100 were treated with the same volume of 1 × RNAse-free TE buffer. Fluorescence signal (ex. 485 nm, em. 520 nm) was determined using a Jasco FP8300 Spectrofluorometer with a microwell plate reader (JASCO Benelux BV, De Meern, The Netherlands).

### 2.5. Stability of Lipid Nanoparticles in Human Plasma

The stability of LNP was determined by asymmetric flow field flow fractionation (AF4) measurements using the AF2000 separation system (Postnova Analytics, Landsberg, Germany). The system is equipped with a degasser, isocratic pumps, auto samples, fractionation channels, and an in-line DLS detector (Zeta Nano ZS, Malvern Instruments, Malvern, UK). For separation, a FFF channel was used with a 350 μm spacer and a regenerated cellulose membrane with a molecular weight cut-off of 10 kDa. PBS was used as mobile phase.

LNP-RNP [HEPES] and LNP-RNP-HDR [HEPES] or LNP-RNP-HDR [H_2_0] were prepared as described above, with a total lipid concentration of 4.4 mM and RNP concentration of 1.6 μM. In addition, 3 μM HDR template was added to the LNP-RNP-HDR formulation. The LNP formulations were not diluted with PBS as described previously, since high concentrations were needed for the AF4 studies. To verify potential destabilizing effects of blood components on the LNP, the nanoparticles were treated with 20% human plasma (#HMPLCIT, BioIVT, West Sussex, UK) and incubated for 1 h at 37 °C. Subsequently, 20 μL were injected at a flow rate of 0.2 mL/min and focused for 4 min with a crossflow of 1.5 mL/min and a focus flow of 1.8 mL/min. After 1 min transition time, the crossflow was kept consistent at 1.5 mL/min for 5 min before it was decreased with a linear decay of 1 to a final cross-flow of 0.5 mL/min over a span of 25 min. Then, the crossflow was decreased with an exponential decay of 0.3 for 30 min until it reached 0 mL/min, at which it was kept constant for 10 min. During the entire run, the detector flow rate was 0.5 mL/min.

### 2.6. Cell Culture

HEK293T stoplight cells and HEK293T cells with stable EGFP expression were cultured in low-glucose DMEM medium supplemented with 10% fetal bovine serum (FBS), at 37 °C and 5% CO_2_. The cell lines were both graciously gifted by Dr. Olivier de Jong and constructed as described previously, using the lentiviral plasmids containing the gene of interest (Stoplight construct [29] or EGFP [30]) in a pHAGE2-EF1a-IRES-PuroR or pHAGE2-EF1a-IRES-NeoR backbone, respectively. Alongside these lentiviral plasmids, HEK293T cells were transfected with pMD2.G plasmid, and PSPAX2 plasmid (Addgene #12259 and #12260, respectively) at a 2:1:1 ratio for lentiviral production. Lentiviral supernatant was then used to transduce HEK293T cells. To prevent multiple integrations of the fluorescent reporter constructs, HEK293T cells were transduced using an MOI < 0.1 and subsequently cultured and expanded with their respective selection antibiotics. After 2 weeks, cells were sorted using a BD FACSAria III cell sorter (Becton Dickinson, Franklin Lakes, NJ, USA), after which they were further expanded in the presence of selection antibiotics.

For subculturing between experiments, 1 mg/mL Gibco^®^ Geneticin^®^ Selective Antibiotic (G418 sulfate, Fischer Scientific, Landsmeer, The Netherlands) was supplemented. Cell culture plastics were acquired from Greiner Bio-One (Alphen aan de Rijn, The Netherlands).

### 2.7. Gene Editing Efficacy Assays

#### 2.7.1. Stoplight Gene Editing Assay

HEK293T stoplight cells were plated at a density of 3 × 10^5^ cells/cm^2^ on a 96-well black plate (Greiner CellStar #655090). The following day, the cells were treated with 10 μL of LNP-RNP supplemented with 1% antibiotic/antimycotic solution (Sigma-Aldrich, Zwijndrecht, The Netherlands). Cells were washed after 24 h with 100 μL of low-glucose DMEM medium supplemented with 10% FBS and 1% antibiotic/antimycotic solution. The cells were incubated for another 24 h at 37 °C and 5% CO_2_. Following this, the cells were treated with 2 μg/mL Hoechst 33342 in complete cell culture medium for 15 min and imaged using a Yokogawa CV7000 Confocal Microscope (Yokogawa Corporation, Tokyo, Japan). The fluorescence image analysis was performed with the Columbus Software (Perkin Elmer, version 2.7.1), of which the analysis workflow is shown in Appendix A. Gene editing efficiency was defined as the number of cells expressing EGFP divided by the number of cells expressing mCherry, as described previously [29]. LNP formulations were compared to a positive control, consisting of RNP delivered using ProDeliverIN CRISPR (Oz Biosciences, San Diego, CA, USA), as specified by the manufacturer, except that a 3.3 μL:1 μg ratio of reagent to protein was used. 

#### 2.7.2. EGFP–BFP Mutation Assay

The mutation of the EGFP signal to BFP as a measure of gene correction was based on the work of Glaser et al. [31]. Briefly, HEK293T-EGFP cells were seeded at a density of 3 × 10^5^ cells/cm^2^ in an appropriate cell culture plate. The following day, medium was supplemented with 1% antibiotic/antimycotic solution and LNP formulations were added to each well, containing a varied concentration of RNP, HDR template, and lipid concentrations. As a positive control, ProDeliverIN CRISPR was used to deliver the RNP and the HDR template in a molar ratio of 15:15:28.5 nM. Cells were washed after 24 h with fresh medium and incubated for two days. Subsequently, they were passaged and expanded for two additional days, leading to a total of five days incubation after transfection. Cells were subsequently harvested, washed twice with PBS, fixed in 1% paraformaldehyde, and transferred to a BD Falcon U-bottom 96-well plate (Becton Dickinson, Franklin Lakes, NJ, USA). 

Cell fluorescence was determined by flow cytometry using the BD FACS CANTO II (Becton Dickinson, Franklin Lakes, NJ, USA). BFP was measured using the Pacific Blue channel of the flow cytometer, while EGFP fluorescence was determined in the FITC channel. Data was analyzed with the Flowlogic software (Inivai Technologies, Mentone, Australia, version 7.3). Gene knock-out was defined as a loss in green fluorescent signal, whereas gene correction was defined as a gain in blue fluorescent signal. The gene editing efficiency was determined by the population negative for EGFP and BFP, indicating gene knock-out, as well as the population positive for blue fluorescence, indicating HDR correction using the specified template. A plasmid encoding this BFP plasmid is given in Appendix A, and was acquired from Twist Bioscience (San Francisco, CA, USA). The gating strategy and model validation are presented in Appendix A.

### 2.8. T7 Endonuclease Assay 

To validate the functional gene-editing readouts, a T7 endonuclease I (T7E1) assay was performed. Genomic DNA was extracted from HEK293T stoplight cells and HEK293T-EGFP cells 2 or 5 days after the transfection with LNP-RNP and LNP-RNP-HDR, respectively, using the PureLink Genomic DNA Mini Kit (Thermo Fisher, Landsmeer, The Netherlands), following the manufacturer’s instructions. PCR amplification was performed using primers designed specifically for the target locus (Appendix A) using Q5^®^ Hot Start High-Fidelity 2X Master Mix (New England Biolabs, Ipswich, MA, USA). Afterwards, PCR products were purified using a QIAquick PCR Purification kit. The PCR products were denatured at 95 °C for 10 min in the presence of NEBuffer 2 (New England Biolabs, Ipswich, MA, USA) and annealed at −2 °C per second temperature ramp to 85 °C, then, at −0.1 °C per second temperature ramp to 25 °C. Following this, hetero-duplexed sequences were incubated with 5U T7E1 enzyme (New England Biolabs, Ipswich, UK) at 37 °C for 18 min to achieve digestion of mismatched DNA. 

## 3. Results

### 3.1. SpCas9 Production, Characterization and Stability in Storage

SpCas9 was recombinantly produced by transforming the LPS-free ClearColi™ BL21 strain with plasmid pET15_SpCas9_NLS_His (Addgene #62731). The elution chromatogram of SpCas9, given in Appendix A, shows that the principal protein component elutes at 250 mM imidazole. To study the long-term stability of in-house produced SpCas9, purified SpCas9 from a representative batch was snap frozen in liquid nitrogen and stored in aliquots at −80 °C until needed for analysis of protein size, activity, and for use in LNP formulations. As shown in Figure 1A, the SpCas9 protein appeared as a clear band on SDS-PAGE at the expected molecular weight of 160 kDa. The relative peak area of the principal SpCas9 band, calculated by SDS-PAGE densitometry, did not deteriorate over time, as shown in Figure 1B (gel excerpts underlying this graph are given in Appendix A). SpCas9, furthermore, proved to be active at introducing a targeted double strand break in plasmid DNA only when complexed with the cognate sgRNA, as seen in agarose gel electrophoresis in Figure 1C. This activity was retained over time, as an activity digest after 12 months of storage showed similarly high SpCas9 activity. The activity did not differ significantly from the positive commercial control for each assay performed over time (Appendix A). These results show that the recombinant SpCas9, produced and stored with these methods and conditions, was active and stable at least for one year. This recombinant SpCas9 was used in subsequent formulation and gene editing studies. 

### 3.2. Characterization and Efficacy of LNP Formulations for Gene Knock-Out (LNP-RNP)

Since pH and ionic strength may influence Cas9 RNP activity as well as RNP complexation during LNP preparation, different LNP formulations for gene knock-out were prepared by varying buffer composition during complexation of RNP with lipids, as well as the total amount of DOTAP in the final LNP-RNP formulations (Figure 2). LNP consistently showed a particle size between 100 nm and 200 nm and a PDI below 0.2, as well as a 𝜁-potential between −5 and −20 mV (Figure 2B,C). Interestingly, the LNP-RNP formulation prepared with nuclease-free water in the complexation phase and containing DOTAP 5 mole% seems to result in a high average particle size and polydispersity index (~1000 nm, PDI 0.8), suggesting this formulation is colloidally unstable, leading to LNP aggregation. A larger polydispersity index was additionally determined for LNP-RNP formulated in nuclease-free water with DOTAP 2 mole%. Quantification of the amount of SpCas9 protein and sgRNA associated with the LNP was done with HPLC and Quant-iTTM RiboGreen^®^ RNA assay, resulting in complexation efficiencies of 63.7% and 68.6% (formulation: DOTAP 5 mole%, 50 mM HEPES buffer for RNP complexation), respectively (Appendix A and S8). As RNP is a 1:1 complex of sgRNA to SpCas9 protein a similar complexation efficiency to lipid nanoparticles is expected, as validated by studying both SpCas9 and sgRNA. Thus, complexation of SpCas9 was used in a further study to compare the effect of formulation buffer on RNP complexation in LNP and interestingly no differences could be detected (Appendix A).

To determine LNP-RNP stability under near-physiological conditions, AF4 was applied to detect intact LNP and measure its average size distribution when incubated in 5× diluted human plasma. The formulations tested during the AF4 studies were LNP-RNP [HEPES], containing DOTAP 0 and 5 mole%. Depicted in Figure 2 are fractograms detected by in-line DLS detectors (Figure 2D,E). LNP show a retention time around 40 min. The peaks on the DLS fractograms of nanoparticles incubated with plasma over the range of the retention times between 10 and 20 min are likely to be plasma proteins, suggested by an overlay of the chromatogram of 20% human plasma (Appendix A). LNP-RNP particles show a significantly higher derived count rate after incubation with plasma (Figure 2D). These results indicate that these LNP do interact with the plasma components, suggesting formation of a protein corona on the surface of the LNP [32]. Based on these findings on particle size, RNP-lipid complexation efficiency and stability, the particles were deemed suitably stable and monodisperse to be tested on reporter cell lines for their gene editing efficiencies.

### 3.3. Determination of Gene Knock Out Efficiency of Different LNP-RNP Formulations

LNP were applied to the HEK293T stoplight cell line to determine functional delivery of RNP. These cells constitutively express mCherry and, upon introduction of a +1 or +2 frameshift targeted by CRISPR-Cas9 downstream of the mCherry coding sequence, co-expression of EGFP is induced [29]. The influence of buffer composition during RNP formation was first assessed, as acidic buffers were shown to be detrimental in past reports [17,23]. Based on EGFP expression percentages, RNP formed in 50 mM HEPES buffer (pH 7.4) or nuclease-free water resulted in much higher gene editing in comparison to citrate or PBS buffer (Figure 3A,C). This was confirmed at the genetic level using the T7E1 assay and TIDE analysis (Figure 3E and Appendix A). An acidic environment clearly has a negative effect on RNP and LNP formation in accordance with the literature [23]. Contrary to these findings, however, limited editing activity was observed in PBS, which is a physiological buffer system. An in vitro activity assay was performed to investigate these effects further. These assays showed that complexation in PBS and citrate leads to irreversible inactivation of the RNP at a DNA-cleavage level (Figure 3B and Appendix A). In contrast, RNP mixed at different NaCl concentrations (up to 1 M) did not lose activity (Appendix A). Taken together these findings indicate that pH or ionic strength alone do not account for the loss of Cas9 activity in the formulations. 

The gene knock-out efficiencies determined by flow cytometry were consistently lower than those determined by image analysis (Appendix A). The higher values obtained with image analysis can be explained by false positives due to difficulties in segmenting individual cells in highly confluent cell images. Nonetheless, flow cytometry confirmed that complexation of the RNP and LNP in HEPES buffer or nuclease-free water are the preferred complexation conditions. As LNP-RNP formulations still have approximately 30–40% of free RNP that was not removed prior to transfection, LNP-RNP transfection efficiencies were compared before and after dialysis overnight against 1 × HBS using a 300 kDa MWCO dialysis membrane to remove free RNP. No difference in gene knock-out efficiency was observed (Appendix A), indicating that gene editing was primarily caused by the RNP complexed to LNP.

A three-way ANOVA was performed to statistically determine the effect of formulation conditions, experimental repeat, and molar ratio of DOTAP on gene knock-out efficiency. Based on the statistical analysis, the LNP-RNP formulation using nuclease-free water resulted in significantly higher gene editing outcomes as compared with those prepared in HEPES buffer (Appendix A). This result depended on the molar ratio of DOTAP used during nanoparticle formulation as well (Appendix A), indicating that RNP and LNP complexation in HEPES buffer requires higher mole% of DOTAP than in water. The statistical analysis, however, does show batch variation from one experiment to another, especially between formulations with HEPES buffer. 

Dose-dependent gene knock-out was studied with two formulations complexed in 50 mM HEPES buffer pH 7.4 and with LNP containing DOTAP 0 or 5 mol% (Figure 3D). From these results, the concentration to reach 50% of the effect (EC_50_) was calculated as a measure of gene knock-out efficiency by fitting a dose-response curve (agonist vs. response) using GraphPad PRISM version 9.1 (r^2^ for LNP-RNP 0% of DOTAP = 0.98, r^2^ for LNP-RNP 5% of DOTAP = 0.99, and r^2^ for ProDeliverIN RNP = 0.93). The LNP formulated with DOTAP 0 mole% have a higher EC_50_ value (0.8 nM) than the formulation with DOTAP 5 mole% (0.2 nM). In comparison, the fit led to an EC_50_ value of 1 nM for the ProDeliverIN positive control. In conclusion, therefore, LNP-RNP with DOTAP 5 mole% formulated in HEPES buffer seems to be the best performing nanoparticle for gene knock-out. Incubation of HEK293T stoplight cells with LNP-RNP did not result in any cytotoxicity at an RNP concentration around 7.7 nM (Appendix A). Incubation of cells with 15 nM of LNP-RNP did result in a lower absolute number of cells (Appendix A).

### 3.4. Characterization of LNP Formulations for Gene Correction (LNP-RNP-HDR)

The LNP formulations additionally containing a single stranded DNA template for HDR-mediated gene correction were optimized using a similar rationale as the LNP-RNP formulations. Water and HEPES buffer at pH 7.4 were selected as primary formulation conditions following the LNP-RNP screening. Further variables were molar ratio of RNP to HDR template, and mole% of DOTAP in the LNP composition. To determine whether ssDNA HDR template had an effect on size and ζ-potential, these values were determined for formulations prepared in HEPES buffer, as differences amongst formulation conditions were not expected as shown in Figure 2. Their characteristics were similar to those found for LNP-RNP (Figure 4A,B), except for the formulation with a 1:1 ratio RNP:HDR, which resulted in a higher polydispersity index. The ζ-potential of these particles was, interestingly, similar to that of the LNP-RNP particles, even though more anionic charges were added to the formulation (up to 10-fold molar excess of template DNA as compared with RNP). 

The plasma interaction of these particles was additionally tested using AF4. The results of the LNP-RNP-HDR particle formulated in nuclease-free water, remarkably, do not show a shift in retention time, as opposed to LNP-RNP (Figure 4C,D). Interestingly, the increased count rate after plasma incubation is less pronounced in particles entrapping HDR template. Moreover, the particles additionally entrapping an HDR template do not change in size (Figure 4D) [33,34]. 

### 3.5. Determination of Gene Correction Efficiency of Different LNP-RNP-HDR Formulations

LNP-RNP-HDR were tested for their gene editing efficacy on HEK293T cells with constitutive EGFP expression. The loss in EGFP indicates gene knock-out, while a gain in the blue signal indicates gene correction (Appendix A). Several concentrations of HDR template were screened (Figure 5A,B), as well as DOTAP percentages. Leaving out DOTAP from the formulation led to a significant reduction in the efficiency of gene editing (Appendix A). The formulation that yielded the highest gene correction efficacy was the LNP prepared in water, which contained DOTAP 0.25 mole% at a 1:2 molar ratio of RNP to HDR template. This formulation yielded a gene correction efficacy of 11.4% of the total cell population, as well as a gene knock-out efficacy of 59.6% of the cells at a final RNP concentration of 7.7 nM. For the LNP formed in HEPES buffer, the overall gene correction efficacies were lower. The percentage of HDR events within the total gene editing outcomes is given in Figure 5B. This percentage is consistently higher for particles complexed in water as compared with HEPES buffer, which indicates that the particles formulated in water were overall more suited for HDR. Another trend is that addition of higher relative concentrations of HDR template is associated with lower gene editing. 

A dose-escalation study was performed for LNP-RNP-HDR formulations prepared in water or HEPES buffer with DOTAP 0.25 mole% and a 1:2 ratio of RNP:HDR template, which performed well in the screening. The dose-dependent toxicity of these formulations after one day was assessed by the MTS assay (Figure 5C). Cell viability decreased slightly over the concentration range but stayed above 90% along the whole concentration range for both formulations. The dose-dependent efficacy was determined by fitting a dose-response curve (agonist vs. response) using Graphpad PRISM version 9.1 for both gene correction (r^2^ for LNP-RNP-HDR [H_2_O] = 0.96 and r^2^ for LNP-RNP-HDR [HEPES] = 0.79) and gene knock-out (r^2^ for LNP-RNP-HDR [H_2_O] = 0.97 and r^2^ for LNP-RNP-HDR [HEPES] = 0.86). These curves showed that formulations prepared in water exhibited a lower EC_50_ (7 nM) for gene correction as compared with the particles prepared in HEPES buffer (47 nM). For gene knock-out, the EC_50_ was lower for all conditions, but the same trend was observed where the water particles showed a lower EC_50_ (1 nM) than HEPES particles (10 nM) (Figure 5D). Gene editing was additionally confirmed by the T7E1 assay (Appendix A), indicating that cells in this population contained insertions or deletions in their genome. These data combined showed that LNP-RNP-HDR formulated in water reached a gene correction efficacy of 19.2% at a concentration of 15 nM RNP with good cytocompatibility (95% cell viability).

## 4. Discussion

We based our formulations on previous literature regarding LNP for mRNA delivery as a starting point [23,35,36]. These use ionizable lipids to simultaneously reduce toxicity, as well as facilitate nucleic acid entrapment and endosomal escape in target cells. Our findings support previous reports, showing that for complexation of RNP, the buffer during RNP and lipid nanoparticle complexation and the inclusion of cationic lipid DOTAP are necessary for stable particles [23]. Moreover, the resulting LNP formulations were biocompatible as highlighted in Figure 5C and Appendix A. We, however, further optimized these conditions for additional complexation of HDR template DNA. Buffer composition during RNP complexation played a major role on its downstream effect on cells. This seems not to be due to Cas9 encapsulation (Appendix A), but rather the Cas9 bioactivity as shown on in vitro gel digests (Appendix A). Whereas the RNP formed in water or HEPES was active, the RNP formed in citrate was not. Citrate, in particular, was tested, as it is used for lipid ionization in reported LNP formulations in the past. To our surprise, in vitro activity of SpCas9 RNP in PBS was severely reduced as well. This suggests that the inhibitory effect is not due to pH or ionic strength during complexation, but rather a specific buffer ion interaction. We showed that HEPES, for example, did yield active RNP in our particles. Further investigation of this effect may reveal buffer incompatibilities of Cas9 RNP.

Another interesting finding is the negative ζ-potential. An explanation for the observed negative ζ-potential could be adsorption of excess RNP to the surface of the LNP [37]. Interestingly, the addition of HDR template does not seem to change particle size or shift the ζ-potential further toward negative, indicating that these are not surface bound (Figure 4A,B).

The efficacy of our optimized particles is in line with the existing literature. Efficiency in gene editing seems to saturate around a concentration of 5–10 nM RNP, thus, higher concentrations of particles would not be required (Figure 3D). In comparison, Suzuki et al. showed editing saturation at 1 nM with their lipid nanoparticles, however, formulation conditions were not comparable to the conditions reported in this study [17]. It is interesting to also note that the incidence of NHEJ-based gene knock-out is more efficient in these formulations than HDR-based gene correction. This ratio, indicated in Figure 5D, needs significant improvement before HDR can be considered for clinical application.

Finally, the AF4 studies show interesting insights with respect to particle stability and potential protein corona formation in the presence of human plasma. More elaborate studies would need to be performed to verify the nature and content of such a protein corona and possibly specify which plasma proteins accumulate on the surface of the particles. Such investigations would be relevant, as a protein corona could mediate specific in vivo localization of the LNP, for example, the adsorption of apolipoprotein E to the surface of LNP results in hepatocyte-specific uptake [33,34]. Incubation with plasma shifts the retention time of LNP formulated without a ssDNA HDR template to a slightly earlier retention time (Figure 2D, Figure 4C and Appendix A). This indicates a change in the particle morphology due to interaction with plasma, which is worth investigating further, and indicates that the HDR template may have a positive influence on particle stability [38]. Previous studies have shown that, in fact, additional anionic charges favor RNP stability in formulations, resulting in better gene editing efficiencies on cells after delivery via electroporation. In any case, these results suggest that the particles are stable for in vivo applications and, thus, warrant further in vivo experimental studies.

## 5. Conclusions

In this study, we set out to find optimized formulation conditions for LNP containing SpCas9 RNP, with and without HDR template. Our main findings are as follows: 

Preparing RNP for formulation in nuclease-free water or HEPES buffer yielded superior gene editing results as compared with PBS or citrate buffer, due to inadequate formation of an active RNP complex in the latter two buffers. There was no marked difference in encapsulation efficiency of Cas9 between these tested systems. Incorporation of DOTAP in the LNP-RNP formulation was associated with a high gene-editing efficacy overall, while for LNP-RNP-HDR, a lower concentration was optimal. High gene knock-out efficacies above 80% were achieved for LNP-RNP prepared in HEPES buffer, with DOTAP 5 mole%, with a clear dose-dependent relationship. As a highlighted result, 20% gene correction efficacy was achieved with LNP-RNP-HDR formulated in nuclease-free water, DOTAP 0.25 mole%, and a 2:1 ratio of HDR template to RNP, with a clear dose-dependent relationship as well, and high cell viability (>90%). 

Moreover, we demonstrated that these LNP formulations remained colloidally stable in the presence of human plasma; however, changes in scattering intensity and average size were detected, which might indicate formation of a protein corona on the particle’s surface. Additionally, we provide a protocol for in-house production, purification, and long-term storage of the SpCas9 protein, which can be stored for at least a year at −80 °C without loss of activity. These findings contribute to understand the necessity of optimal formulation conditions to create LNP for direct in vivo delivery of CRISPR-Cas9 components.

## Figures and Tables

**Figure 1 pharmaceutics-14-00213-f001:**
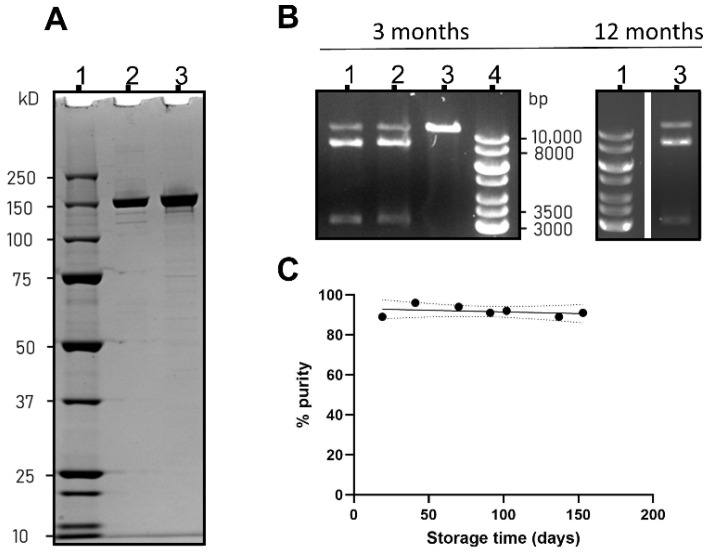
SpCas9 characterization after purification and extended storage: (**A**) SDS-PAGE gel of the purified recombinant SpCas9 ((**1**) PageRuler Plus prestained protein ladder; (**2**) positive control SpCas9 acquired from Sigma-Aldrich; (**3**) in-house produced SpCas9); (**B**) Relative density of the 160 kDa protein band on the SDS-PAGE gels over time, defined as percentage purity; (**C**) Activity of the SpCas9 protein (lane 1 left gel, lane 3 right gel) as compared with a commercial sample (lane 2, left gel) and a negative control (lane 3, left gel). Gene ruler 1kB ladder (lane 4 left gel, lane 1 right gel) was used for determining the size of the DNA fragments. The activity is shown for SpCas9 after 3 months and 1 year in storage.

**Figure 2 pharmaceutics-14-00213-f002:**
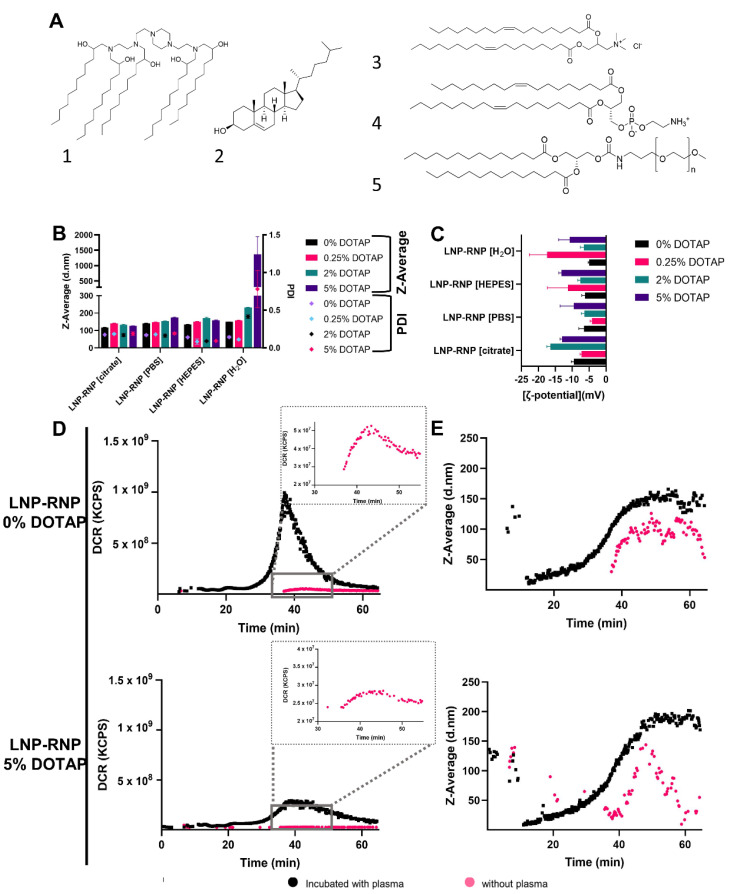
LNP characterization and plasma stability: (**A**) Chemical structures of the LNP components in the formulations ((**1**) C12-200; (**2**) cholesterol; (**3**) DOTAP; (**4**) DOPE; (**5**) PEG-DMG); (**B**) and (**C**) LNP-RNP characteristics screened for varying DOTAP concentrations and complexation buffers, (**B**) average particle size and PDI in PBS as determined by DLS (measured in triplicate) and (**C**) ζ -potential of these formulations in 10 mM HEPES buffer pH 7.4 (measured in triplicate). Two of these formulations were further characterized on stability in plasma (AF4); (**D)** and (**E**) AF4 fractograms recorded by DLS detector showing the derived count rate (**D**) and particle size (**E**) for LNP-RNP formulated in HEPES buffer with DOTAP 0 and 5 mole%. Inserts show a zoomed-in version of the samples measured without plasma. Detector flow was set to 0.5 mL/min.

**Figure 3 pharmaceutics-14-00213-f003:**
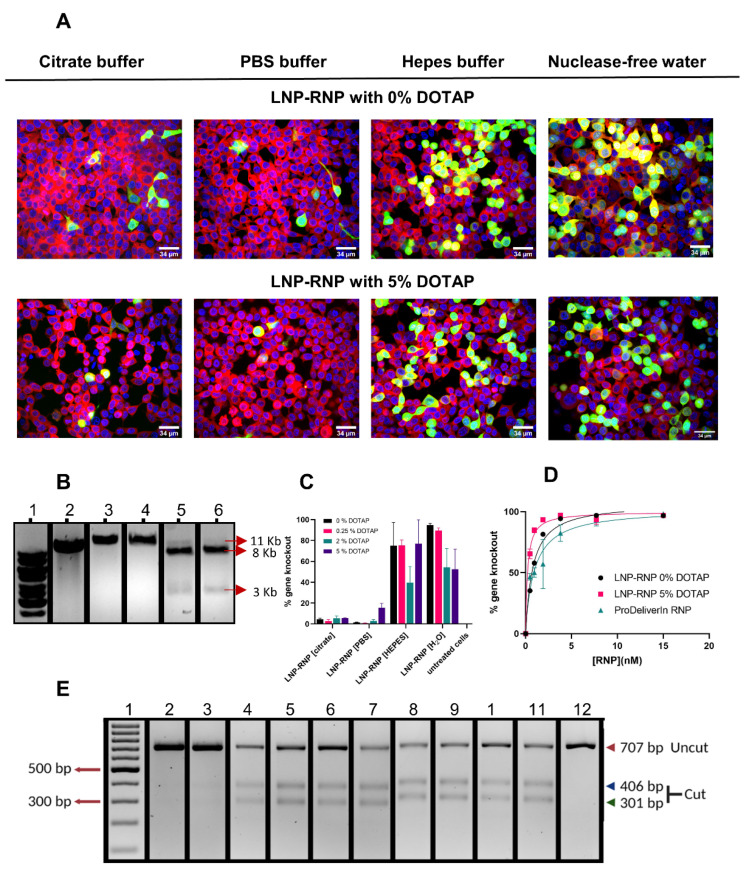
Determination of gene knock-out efficiency in HEK293T stoplight cells: (**A**) Confocal microscopy images (60 ×) of HEK293T stoplight cells after treatment with different LNP formulations at a RNP concentration of 7.7 nM (RNP were complexed in different conditions, i.e., 100 mM citrate buffer, PBS buffer, 50 mM Hepes buffer, and nuclease-free water). Red represents mCherry, green represents EGFP (Cas9 gene editing), and blue represents Hoechst (nucleus). Scale bar 34 μm. Images were optimized on ImageJ in brightness and contrast for each channel, respectively; (**B**) Cas9 activity in vitro using the same buffers as in (**A**) during RNP complexation. Uncut (11 kB) and cut (8kB and 3kB) DNA are highlighted by arrows. (**1**) Generuler 1 kB DNA ladder; (**2**) untreated DNA; (**3**–**6**) RNP complexed in citrate (**3**), PBS (**4**), HEPES (**5**), or water (**6**); (**C**) gene knock-out efficiencies for different LNP formulations (with final RNP concentration 7.7 nM) determined by confocal image analysis using Columbus^®^ software (tested in triplicate); (**D**) dose-dependent gene knock-out efficiencies of two selected LNP-RNP formulations (0% DOTAP and 5% DOTAP, 50 mM HEPES buffer) as compared with the commercial transfection agent, ProDeliverIN (tested in duplicate); (**E**) T7E1 digests performed on the same samples and ordered as in panel (**C**). (**1**) DNA ladder; (**2**) LNP-RNP containing DOTAP 5 mole%, prepared in 100 mM citrate buffer; (**3**) LNP-RNP containing DOTAP 5 mole%, prepared in PBS; (**4**–**7**) LNP-RNP prepared in 50 mM HEPES buffer with DOTAP 0, 0.25, 2 and 5 mole%, respectively; (**8**–**11**) LNP-RNP prepared in water with DOTAP 0, 0.25, 2, and 5 mole%, respectively; (**12**) negative control. The unedited gel is provided in Appendix A as the order of the lanes was changed for clarity within this figure.

**Figure 4 pharmaceutics-14-00213-f004:**
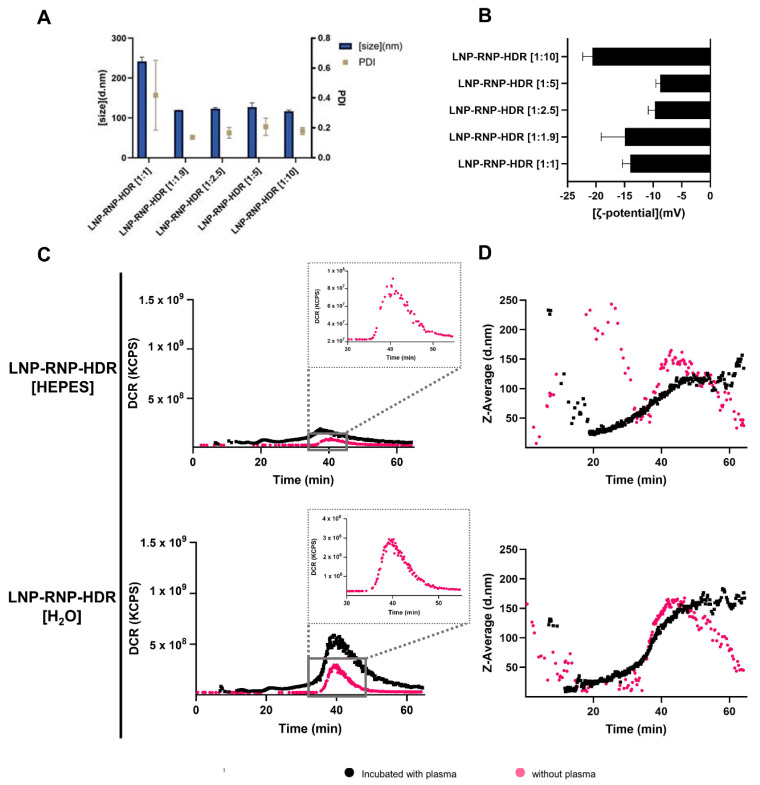
Characterization of LNP-RNP-HDR formulations: (**A**,**B**) Representative LNP-RNP-HDR characteristics screened for varying HDR template concentrations (in molar ratios as compared with RNP) at a fixed complexation buffer (50 mM HEPES) and at a fixed lipid composition (DOTAP 0.25 mole%). (**A**) Average particle size and PDI as determined by DLS (measure in triplicate) and (**B**) ζ -potential of these formulations in 10 mM HEPES buffer pH 7.4 (measured in triplicate); (**C**) AF4 fractograms recorded by DLS detector showing the derived count rate (DCR) of LNP-RNP-HDR formulations at a fixed HDR template concentration (1:1.9 molar ratio) and DOTAP concentration (0.25 mole%) in varying complexation buffers, with and without plasma incubation; (**D**) AF4 fractograms recorded by DLS detector of particle size for LNP-RNP-HDR (same formulations as in (**C**)). Detector flow was set to 0.5 mL/min.

**Figure 5 pharmaceutics-14-00213-f005:**
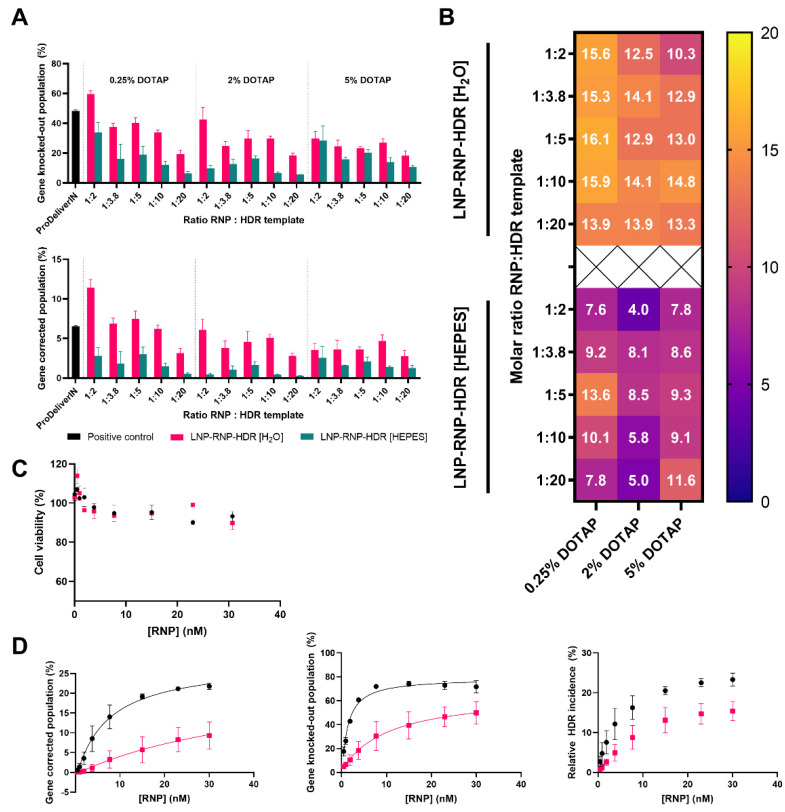
(**A**) Formulation optimization to achieve gene correction using LNP at an RNP concentration of 7.7 nM, with varying molar ratios of RNP/HDR template and percentages of DOTAP in the lipid composition (tested in triplicate). Complexation of RNP and lipids was performed in water or HEPES prior to transfection. The concentration of DOTAP and template DNA was varied; (**B**) heatmap representation of the relative gene correction ratio (percentage incidence as compared with the sum of outcomes) within the gene-edited populations of Figure 5A; (**C**) MTS cell viability of a dose range of the best performing formulations formed in HEPES buffer (pink) or water (black), containing DOTAP 0.25 mole% and a 1:2 molar ratio of HDR template to RNP (tested in duplicate); (**D**) dose escalation study performed with the same formulations in (**C**) (pooled data from 2 batches, pink represents HEPES buffer and black represents water), represented for the gene correction, gene knock-out, and relative incidence of HDR as percentages within the gene-edited population (tested in duplicate).

## Data Availability

Supporting data to the research can be found in the Appendix A of this manuscript.

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
