# Peer review of "Impact of Formulation Conditions on Lipid Nanoparticle Characteristics and Functional Delivery of CRISPR RNP for Gene Knock-Out and Correction"

_pharmaceutics, 2022, doi:10.3390/pharmaceutics14010213_

Round 1

Reviewer 1 Report

The manuscript “Impact of formulation conditions on lipid nanoparticle characteristics and functional delivery of CRISPR RNP for gene knock-out and correction” describe the preparation of gene delivery systems based on lipid nanoparticles (LNP) and ribonucleoprotein (RNP). The authors investigated the effect of different formulation conditions on the particle’s characteristics as well as on their efficiency.

The article is interesting and well written, therefore I suggest to be published. I have only minor comments in order to improve the manuscript.  

  • According me the abstract does not well describe the article. It seems more like introduction. The investigations, experiments and findings are only weakly mentioned. The manuscript will attract more audience if the abstract is written more clearly.
  • Too much abbreviations are used and that confuse the reader……Some of them are not clarified such as DOPE, DOTAP, PEG-DMG etc. Although some of them are popular they should be
  • Dynamic and electrophoretic light scattering is poorly described in the experimental section. More information about the measurement conditions should be specified (T°, pH etc.).
  • Check the font size and style on lines 432-433

Reviewer 2 Report

The approach of research work entitled “Impact of formulation conditions on lipid nanoparticle characteristics and functional delivery of CRISPR RNP for gene knock-out and correction” submitted to the “Pharmaceutics” journal seems good and scientific rich content, however prior acceptance, some minor suggestions have been incorporate. 

  1. Author prepared the lipid nanoparticles, it is not clear which method they used with proper citation, else they are reported first time.
  2. The very few characterization techniques except particle size, PDI only and formulation aspects have been covered, so far.
  3. Morphological studies need to be incorporated for size and shape.
  4. On what basis author selected the lipids, it is also not added.

Reviewer 3 Report

The manuscript “Impact of formulation conditions on lipid nanoparticle characteristics and functional delivery of CRISPR RNP for gene 3 knock-out and correction” by Walther et al focuses on optimising formulation conditions to produce efficient gene loaded lipid nanoparticle. I found the manuscript interesting, of a robust experimental design and valid rational.  All methods are validated, and results are appropriately presented. I recommend publications after the following corrections:

  • Abstract must be rewritten, the current looks loke a literature or an abstract for a review rather than research article
  • All abbreviations (especially in the methods and materials) must be written in full name once mentioned the first time.
  • Minor English editing is needed
  • SEM OR TEM of the lipid nanoparticle

Reviewer 4 Report

Title of work to be redefine to correspond to the work incoporated.

More characterization/evaluation of nanoparticle to be included e.g TEM, EE etc.

In vivo (pharmacokinetic or pharmacodynamic performance) could autheticate the proof of concept.

Txoicology, in vivo stability and regulatory issue should also be addressed.

Round 2

Reviewer 4 Report

The author has adressed all the concerns.